# Tannic Acid-Modified Silver Nanoparticles in Conjunction with Contact Lens Solutions Are Useful for Progress against the Adhesion of *Acanthamoeba* spp. to Contact Lenses

**DOI:** 10.3390/microorganisms10061076

**Published:** 2022-05-24

**Authors:** Marcin Padzik, Lidia Chomicz, Julita Bluszcz, Karolina Maleszewska, Jaroslaw Grobelny, David Bruce Conn, Edyta B. Hendiger

**Affiliations:** 1Parasitology Laboratory, Department of Medical Biology, Medical University of Warsaw, Litewska 14/16, 00-575 Warsaw, Poland; lidia.chomicz@wum.edu.pl (L.C.); julitabluszcz@gmail.com (J.B.); karolinamela12@gmail.com (K.M.); edyta.hendiger@wum.edu.pl (E.B.H.); 2Department of Materials Technology and Chemistry, Faculty of Chemistry, University of Lodz, 163 Pomorska Street, 90-236 Lodz, Poland; jaroslaw.grobelny@chemia.uni.lodz.pl; 3Department of Invertebrate Zoology, Museum of Comparative Zoology, Harvard University, Cambridge, MA 02138, USA; dconn@lsdiv.harvard.edu or; 4One Health Center, School of Mathematical and Natural Sciences, Berry College, Mount Berry, GA 30149, USA

**Keywords:** *Acanthamoeba* spp., *Acanthamoeba* keratitis, contact lenses, contact lens solutions, nanoparticles, tannic acid, anti-adhesion potential

## Abstract

*Acanthamoeba* spp. are amphizoic amoebae that are widely distributed in the environment and capable of entering the human body. They can cause pathogenic effects in different tissues and organs, including *Acanthamoeba* keratitis (AK), which may result in a loss of visual acuity and blindness. The diagnostics, treatment, and prevention of AK are still challenging. More than 90% of AK cases are related to the irresponsible wearing of contact lenses. However, even proper lens care does not sufficiently protect against this eye disease, as amoebae have been also found in contact lens solutions and contact lens storage containers. The adhesion of the amoebae to the contact lens surface is the first step in developing this eye infection. To limit the incidence of AK, it is important to enhance the anti-adhesive activity of the most popular contact lens solutions. Currently, silver nanoparticles (AgNPs) are used as modern antimicrobial agents. Their effectiveness against *Acanthamoeba* spp., especially with the addition of plant metabolites, such as tannic acid, has been confirmed. Here, we present the results of our further studies on the anti-adhesion potential of tannic acid-modified silver nanoparticles (AgTANPs) in combination with selected contact lens solutions against *Acanthamoeba* spp. on four groups of contact lenses. The obtained results showed an increased anti-adhesion activity of contact lens solutions in conjunction with AgTANPs with a limited cytotoxicity effect compared to contact lens solutions acting alone. This may provide a benefit in improving the prevention of amoebae eye infections. However, there is still a need for further studies on different pathogenic strains of *Acanthamoeba* in order to assess the adhesion of the cysts to the contact lens surface and to reveal a more comprehensive picture of the activity of AgTANPs and contact lens solutions.

## 1. Introduction

The number of people wearing contact lenses is constantly increasing, along with corneal infections caused by viruses, bacteria, and fungi resulting from the improper use and care of contact lenses. The worldwide *Acanthamoeba* keratitis (AK) rate has been also increasing over recent years among contact lens wearers [1]. Amoebae of the *Acanthamoeba* genus are detected worldwide in both wet and dry environments. Their presence has been confirmed in contact lens solutions and contact lens storage cases. The adhesion of amoebae to the contact lens surface is the first step in development of AK. Ninety percent of confirmed AK cases involve contact lens wearers. Statistics show that 70 contact lens wearers per million will develop AK. The late diagnosis of infection can lead to complete loss of vision [1,2,3,4,5,6,7].

The symptoms of AK are generally non-specific and are similar to those of bacterial, fungal, or viral keratitis. A ring-shaped infiltrate of the cornea is the most characteristic symptom of AK; however, this occurs in only 50% of cases. The misdiagnosis of AK is common and causes delays in proper treatment [8,9,10]. Medical practice has confirmed that only the initiation of therapy at the early stage of infection can lead to full recovery. Microinjuries and ulcerations, caused by friction between the contact lens and the cornea, are the main factors promoting amoebae adhesion. The amoebae will then penetrate deeper layers of the cornea, finally infiltrating the nerve. The infection is usually unilateral and manifests itself through severe eye pain, lentigines, massive tearing, and drooping eyelids [11,12,13].

The treatment protocol for AK recommended by the Centers for Disease Control and Prevention (CDC) is long-term, toxic, and non-specific. Additionally, it can cause serious side effects, including permanent eye damage [14,15,16]. The drugs typically used and recommended belong to the diamides and biguanides groups; these include polyhexamethylene biguanide (PHMB) and chlorhexidine digluconate. Different treatment regimens are used depending on the stage of the disease. The most promising initial treatment is a combination therapy consisting of chlorhexidine with propamidine, hexamidine, dibromopropamidine, isethionate, aminoglycosides, and neomycin (0.05%), and is more effective than monotherapy. Additionally, antifungal agents, such as miconazole, ketoconazole, fluconazole voriconazole, and clotrimazole, may be used as protective drugs [10,17]. It is recommended to remove the corneal epithelium before starting the treatment. Most antibiotics are not effective against *Acanthamoeba*; however, Siddiqui et al. recommended the use of chloramphenicol [18]. The use of corticosteroids is controversial due to their possible side effects, such as a decreased patient immune response. Glucocorticoids are only used in combination with anti-amoebic therapy in the later stages of treatment. In advanced stages of the disease, a corneal transplant is necessary. After the operation, it is crucial to continue the anti-amoebic treatment for several months [19].

It is known that prevention, including education of contact lens wearers on proper contact lens management, is major factor in reducing the risk of developing AK. Lenses should be used and managed as recommended by the manufacturer and for the length of time specified on the label. Furthermore, proper care, including rinsing contact lenses in designated contact lens solutions, is also extremely important. However, most of contact lens solutions are not effective against *Acanthamoeba* trophozoites and cysts. Therefore, there is still an urgent need to enhance their anti-amoebic and anti-adhesive properties [20,21,22,23,24,25,26,27].

Nanoparticles (NPs), as next-generation agents, are of increasing interest to researchers all over the world. The wide possibilities of their application make them modern antibacterial, antifungal, and antiviral agents [28,29,30,31]. Their activity against protists, such as *Toxoplasma gondii* and *Giardia intestinalis*, has also been shown [32,33]. NPs range in diameter between 1 and 100 nm. They may have a unique shape modifiability that influences their physicochemical properties and allows them to penetrate cells, causing various effects.

The mechanism of action of NPs has not been precisely described yet. Initially, it involves the disruption of cell membrane structures. NPs penetrate the cell interior and produce reactive oxygen species (ROS) that affect respiratory pathway enzymes, stop protein synthesis, and have a destructive effect on DNA replication. NPs can also be used as drug carriers [31,34,35]. Silver nanoparticles (AgNPs) are the most studied NPs thus far, and have been shown to have the best antimicrobial activity. In one study, AgNPs also showed anti-amoebic activity and reduced the adhesion of amoebae to the contact lens surface, with limited cytotoxic effects on the patient [34].

AgNPs combined with plant extracts are even more effective than AgNPs alone. For example, studies performed on AgNPs conjugated with *Jatropha curcus*, *Jathopha gossypifolia*, and *Euphorbia* sp. extracts showed enhanced antimicrobial effects compared to pure AgNPs [36]. Tannic acid (penta-m-digalloyl glucose) is the simplest polyphenolic plant metabolite with proven antioxidant, anticancer, and antimicrobial activity. The properties of tannic acid include the ability to form insoluble complexes with nucleic acids, carbohydrates, proteins, and chelating metal ions [37,38,39,40,41]. Previously, we revealed that tannic acid-modified silver nanoparticles (AgTANPs) had anti-amoebic activity against trophozoites of *Acanthamoeba* clinical isolates: *Acanthamoeba polyphaga* and two other clinical strains of the *Acanthamoeba* T4 genotype. The obtained electron microscopy images showed good penetration of the nanoparticles into the amoebic cells [42]. Here, we attempted to determine and assess the anti-adhesion potential of AgTANPs in combination with selected contact lens solutions against *Acanthamoeba* trophozoites (Neff strain) on four groups of contact lenses (as per FDA classification).

### 1.1. Acanthamoeba Cultivation

For the adhesion assays, *Acanthamoeba castellanii* (Neff strain; ATCC 30010; LG Promochem, Barcelona, Spain) was selected. The axenic cultivation of the *Acanthamoeba* strain was performed in culture tissue flasks at room temperature. The culture medium (peptone yeast glucose (PYG)) contained 0.75% (*w*/*v*) protease peptone, 0.75% (*w*/*v*) yeast extract, and 1.5% (*w*/*v*) glucose with 10 µg of gentamicin mL^−1^ (Biochrom AG, Cultek, Granollers, Barcelona, Spain). The experiment was performed at the Department of Medical Biology, Medical University of Warsaw, Poland. The subculture of the *Acanthamoeba* strain was performed three days prior to the experiment (logarithmic phase of growth) and monitored under a Leica DMIL inverted microscope (Leica, Wetzlar, Germany).

### 1.2. Nanoparticles

Silver nanoparticles (AgNPs) were synthesized by a chemical reduction method using silver nitrate (AgNO_3_; purity 99.999%; Sigma-Aldrich, St Louis, MO, USA). Tannic acid-modified silver nanoparticles (AgTANPs) were prepared by mixing a heated aqueous solution of AgNO_3_ (95.2 g, 0.017%) with an aqueous solution of a tannic acid (0.6 g, 5% C 76H52 O46; Sigma-Aldrich). The long-term stability of the colloidal dispersions of all tested NPs (ζ potential) was measured and confirmed by the electrophoretic light-scattering method with a Zetasizer Nano ZS (model ZEN3500; Malvern Instruments, Worcestershire, UK). The size and shape of the AgTANPs were determined by high-resolution scanning transmission electron microscopy (HR-STEM) as previously described in [34,42]. The well-dispersed nanofluids were appropriately diluted to concentrations ranging between 1.25–10 ppm and used in the assays.

### 1.3. Contact Lens Solutions

The three most popular multipurpose contact lens solutions on the Polish market were investigated: Solo Care Aqua (SCA), Opti-Free (O-F) and ReNu MultiPlus (ReNu). The contact lens solution ingredients are shown in Table 1. All solutions used during the study were obtained from authorized agents (pharmacy), combined with AgTANPs, and used in the *Acanthamoeba* (Neff strain) adhesion assays.

### 1.4. Cytotoxicity

The cytotoxicity assays were performed using PCS-201–010 normal fibroblast cell lines, as described in our previous study [42]. A commercial kit for the investigation of drug-induced cytotoxic effect based on the measurement of lactate dehydrogenase (LDH) activity released to the media (Pierce LDH cytotoxicity assay kits 88953, 88954) was used according to the manufacturer’s protocol. The fibroblasts were incubated with each concentration (ppm) of the AgTANPs separately. To calculate the percent cytotoxicity, absorbance was measured at 490 nm and 680 nm as per the formula below. The cytotoxicity results were promising, and varied from 32.92% at a 10ppm concentration of AgTANPs to 18.84% at a 1.25 ppm concentration of AgTANPs.
%Cytotoxicity = [(EV-ECSC-TCSC) / (TCMC-TCSC)] × 100

EV—Experimental value ECSC—Effector cells spontaneous controlTCSC—Target cells spontaneous controlTCMC—Target cell maximum control

### 1.5. Adhesion to Contact Lenses

Four types of hydrogel contact lenses, as classified according to the US Food and Drug Administration (FDA), were obtained from authorized agents (pharmacy). The contact lenses were placed gently in each well of 24-well microtiter plates with laboratory tweezers and incubated for 90 min with 1000 µL of the *Acanthamoeba* suspension, at a concentration of 10^5^ cells per well. The *Acanthamoeba* trophozoites adhered to the contact lens surface at 24 °C. After that, each lens was transferred with tweezers to a new well and rinsed with saline solution (0.9% NaCl). The number of cells attached to each contact lens was verified under an inverted microscope (OPTA-TECH MW50 with an OPTA-TECH MI5FL 5 MP digital camera). All the lenses were then exposed to 400 µL of selected contact lens solutions mixed with 100 µL of AgTANPs at concentrations of 10, 5, 2.5, and 1.25 ppm (with a different concentration in every well). As a control, the contact lenses were incubated with 400 µL of selected contact lens solution and 100 µL of Milli-Q purified water, instead of NPs. Further incubation continued for either 4 or 6 h depending on the minimum disinfection time recommended by the manufacturer (provided in Table 2). To verify the *Acanthamoeba* trophozoites adhesion, each lens was transferred to a new well, filled with saline solution, and monitored with an inverted microscope (OPTA-TECH MW50 with an OPTA-TECH MI5FL 5 MP digital camera). The adhesion reduction (AR) was calculated using the following formula:*AR* = (*nc* − *nt*)/*nc* × 100%(1)
where *nc* is the number of attached amoebae in the control well and *nt* is the number of attached amoebae in the test well. All experiments were repeated 3 times in triplicate.

### 1.6. Amoebae Adhesion—Control

The adhesion of the amoebae to the contact lenses’ surface varied and depended on the type of contact lens used. The strongest adhesion was obtained after a 90-min incubation of the amoebae suspension with contact lenses from FDA groups 3 and 4. These lenses are made of ionized materials. The observed adhesion was a monolayer, not susceptible to saline rinsing, and regular, in contrast to the adhesion to other lenses. The poorest adhesion was visualized in contact lenses from FDA groups 1 and 2, as shown in Figure 1.

### 1.7. Amoebae Adhesion—Contact Lens Solutions Only

After 4 or 6 h of incubation (depending on the minimum disinfection time recommended by the manufacturer), the anti-adhesion activity of three tested contact lens solutions was verified on four types of contact lenses. The best amoebae adhesion reduction (AR) was demonstrated by O-F on the FDA group 4 contact lenses—up to 36.5%. SCA reduced amoebic adhesion on three types of contact lenses, while O-F and ReNu showed activity on only two FDA groups of contact lenses. However, the overall adhesion reduction in all tested contact lens solutions was not satisfactory. Detailed results are listed in Table 3.

### 1.8. Amoebae Adhesion—Contact Lens Solutions + AgTANPs

The addition of AgTANPs to contact lens solutions in concentrations of 5 ppm and 10 ppm resulted in a significant enhancement of the adhesion reduction in all types of the tested contact lenses compared to the results obtained for contact lens solutions acting alone. In the FDA 1 and FDA 3 groups of contact lenses, the most beneficial dose-dependent effect on adhesion reduction was shown with SCA + AgTANPs. As confirmed in our previous studies [27] and based on the current morphological assessment of the trophozoites (Figure 2), it seems that contact lens solutions in conjunction with AgTANPs influence the cell shape by reducing the number of acanthopodia, thus affecting the adhesion process. In the FDA 2 and FDA 4 groups of contact lenses, the most beneficial dose-dependent effect on adhesion reduction (AR) was shown with O-F + AgTANPs. The worst anti-adhesive effect in the conducted experiment was shown by ReNu + AgTANPs against the FDA 4 group. Detailed results are listed in Table 3.

## 2. Conclusions and Perspectives

The adhesion of *Acanthamoeba* trophozoites to contact lenses and their transmission from the contact lens surface to the corneal epithelium is the first step to AK development [11]. Here, we showed that adhesion to lenses composed of ionized material (FDA groups 3 and 4) was much higher and stronger compared to non-ionized lenses (FDA groups 1 and 2) [27]. The obtained results, which are also supported by those from other authors, suggest that the ionization of contact lens materials may be a significant factor favoring *Acanthamoeba* adhesion [7]. On the other hand, we did not confirm any relationship between the water content in the lenses and the adhesion of amoebae, thus corroborating the results of a previous study performed by Bakay et al. in 2017 [43]. 

To avoid AK, a devastating sight-threatening disease among contact lens users, it is necessary to disinfect contact lenses properly to prohibit amoebae adhesion to their surface. In 2011, the effectiveness of eight universal contact lens solutions, including two types of hydrogen peroxide components and a povidone-iodine-containing solution, was examined to determine their efficacy against *Acanthamoeba* trophozoites and cysts. The obtained results showed a very limited effectiveness of the tested solutions against *Acanthamoeba*. The only effective agent was the povidone-iodine solution [44]. The most popular contact lens solutions, including Solo Care Aqua, Opti-Free, and ReNu MultiPlus, were tested in our previous studies in 2014 and did not show any effectiveness against *Acanthamoeba* trophozoites within the recommended disinfection time [25]. Our results were corroborated by Niyyati, M. et al. in 2018 in their study on *Acanthamoeba* cysts. The only anti-cystic activity was revealed for Opti-Free solution after 6 days of incubation [45]. Similar results were obtained by Hussain R.H.M. et al. and Fears, A.C. et al., who reported that none of their tested multipurpose contact lens solutions showed amoebicidal or cysticidal effects. Moon, E.K. et al. revealed that most multipurpose contact lens solutions made in Korea were ineffective against *Acanthamoeba* trophozoites and cysts. The solutions with cysticidal activity showed increased cytotoxicity to human corneal epithelial cells (>50%). Lakhundi, S., et al., using qualitative assays, revealed that none of the contact lens disinfection solutions studied exhibited cysticidal effects [20,21,22,23]. Here, we showed that the various anti-adhesion effects of the contact lens solutions depended on the FDA group of contact lens used and the type of the contact lens solution tested. In general, the adhesion reduction was unsatisfactory and varied from 4 to 36.5%. None of the tested contact lens solutions reduced amoebae adhesion in all four types of contact lenses. We obtained similar results in our previous study in 2021 [27]. 

The use of nanoparticles (NPs), with their antibacterial, antifungal, antiviral, and antiparasitic properties, is a promising approach to the treatment and prophylaxis of infectious diseases, including AK. However, NPs combined with plant components have been tested mainly for their antibacterial, antiviral, or anticancer activities. Silver nanoparticles (AgNPs) conjugated with the extract of *Oscillatoria limnetica* exhibited strong antibacterial activity against multidrug-resistant bacteria, as well as cytotoxic effects against both a human breast cancer cell line and a human colon cancer cell line [46]. The combined use of AgNPs with *Peganum harmala* L. leaf extract resulted in significant inhibitory effects against clinical isolates of *E. coli* and *S. aureus* [47]. AgNPs conjugated with tannic acid showed antiviral activity in herpes simplex virus type 2 infection [48]. We previously tested AgTANPs in conjunction with selected contact lens solutions and confirmed their enhanced anti-amoebic activity, without any increase in their cytotoxicity to human cells [26]. The cytotoxicity tests were performed on fibroblasts, which are more robust and stable in laboratory *in vitro* conditions than human corneal cells; therefore, the results obtained are more accurate. In addition, the *Acanthamoeba* Neff strain used in our adhesion assays showed more stable population dynamics, which allowed us to perform all tests in the logarithmic growth phase of the trophozoites population. We also showed that pure AgNPs significantly increased the anti-adhesion properties of the selected contact lens solutions [27]. In the present study, we showed that silver nanoparticles in conjunction with tannic acid (AgTANPs) increased the anti-adhesion properties of the tested contact lens solutions even more than pure AgNPs. The addition of 10 ppm of AgTANPs resulted in a significant reduction in amoebae adhesion from 70% to 100% in all tested contact lens solutions and on all tested contact lens FDA groups. These results are very promising in terms of the low cytotoxicity of AgTANPs to human cells at the concentrations used in these assays. However, there are some limitations that should be taken into consideration in future studies. Among them, an assessment of additional pathogenic clinical strains of *Acanthamoeba* and the adhesion of the cysts to the contact lens surface are necessary to see a more complete picture of the activities of the AgTANPs and contact lens solutions.

The global contact lenses market size is currently worth ca. $9.94 billion and is projected to grow to $14.80 billion by 2029 [49]. These data suggest that the worldwide *Acanthamoeba* keratitis (AK) rate will increase accordingly. The ease of amoeba adhesion to the contact lens surface and the questionable anti-amoebic and anti-adhesive efficacy of contact lens solutions presents serious challenges for the prevention of *Aanthamoeba* eye infections. Here, the tested contact lens solutions themselves did not show a satisfactory anti-adhesive effect on *Acanthamoeba* trophozoites in any of the selected types of contact lenses; therefore, they did not sufficiently protect against AK development. The same contact lens solutions used in combination with AgTANPs showed a significantly better anti-adhesive effect, especially at higher concentrations. This effect may provide a benefit in improving the anti-amoebic effectiveness of contact lens solutions and thereby may be useful in improving the prevention of amoebae eye infections. Subsequent assays should be performed to determine and assess the agents influencing the adhesive properties shown by trophozoites and cysts of other *Acanthamoeba* strains. Further comparative attempts may allow us to expand our understanding of this anti-adhesive effect and its potential in minimizing the risk of *Acanthamoeba* spp. diseases in humans.

## Figures and Tables

**Figure 1 microorganisms-10-01076-f001:**
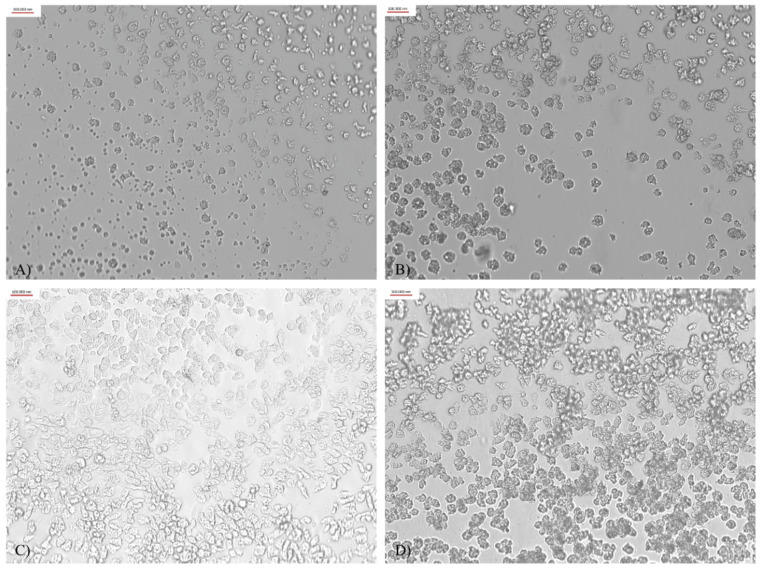
Trophozoite adhesion to the contact lens surface after 90-min incubation (×400 magnification): (**A**) FDA group 1; (**B**) FDA group 2; (**C**) FDA group 3; (**D**) FDA group 4.

**Figure 2 microorganisms-10-01076-f002:**
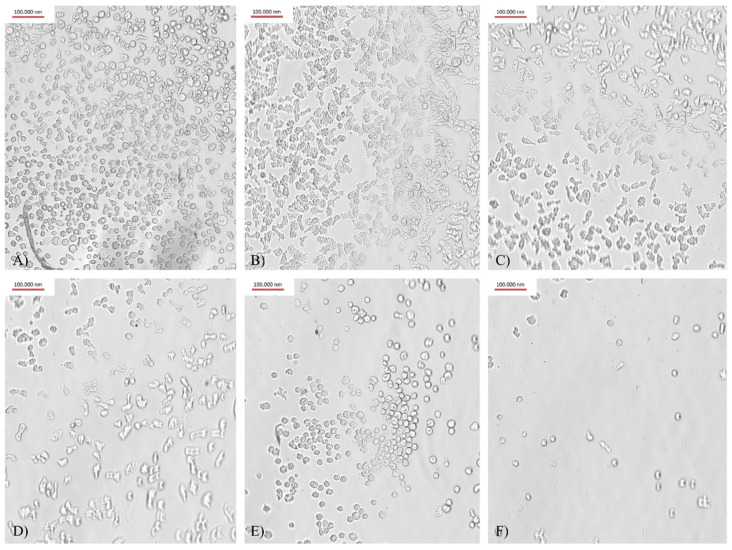
Trophozoites adhered to FDA group 3 contact lenses’ surface after 4 h of incubation (×400 magnification): (**A**) water control; (**B**) SCA solution control; (**C**) SCA + 1.25 ppm AgTANPs; (**D**) SCA + 2.5 ppm AgTANPs; (**E**) SCA + 5 ppm AgTANPs; (**F**) SCA + 10 ppm AgTANPs. The AgTANPs dose-dependent adhesion reduction (AR) in trophozoites is visible in the photos (**C**–**F**).

**Table 1 microorganisms-10-01076-t001:** Composition of the multipurpose contact lens solutions and minimum disinfection times recommended by the manufacturers.

Manufacturer	Solution	Ingredients	Minimum Disinfection Time (h)
Menicon	Solo Care Aqua (SCA)	Polyhexanide (0.0001%), Hydrolock (dexpanthenol, sorbitol), sodium phosphate, tromethamine, poloxamer 407, disodium edetate	4
Alcon	Opti-Free (O-F)	TearGlyde (Tetronic 1304, nonanoyl ethylenediaminetriacetic acid), Polyquad (polyquaternium-1; 0.001%), Aldox (myristamidopropyl dimethylamine; 0.0005%)	6
Bausch + Lomb	ReNu MultiPlus (ReNu)	Hydranate (hydroxyalkylphosphonate; 0.03%), boric acid, edetate disodium, poloxamine (1%), sodium borate, sodium chloride, preserved with Dymed (polyaminopropyl biguanide; 0.0001%)	4

**Table 2 microorganisms-10-01076-t002:** Characterization of selected hydrogel contact lenses as per FDA classification.

FDA Group	Manufacturer	Polymer	Water Content	Ionic
1	Acuvue Oasys 1-Day with Hydraluxe	Senofilcon A	38%	No
2	Focus Dailes All Day Comfort	Nelfilcon A	69%	No
3	Bausch + Lomb PureVision	Balafilcon A	36%	Yes
4	Daily FitViev	Methafilcon A	56%	Yes

**Table 3 microorganisms-10-01076-t003:** Percentage (%) of amoebae adhesion reduction (AR) ± standard deviation (SD) after incubation with tested agents depending on the FDA type of contact lenses used.

	FDA 1	FDA 2	FDA 3	FDA 4
SCA	17.61 ± 3.69	no activity	21.47 ± 16.83	23.47 ± 41.59
SCA + 1.25 ppm AgTANPs	46.31 ± 1.55	26.58 ± 41.44	26.79 ± 26.79	17.33 ± 1.47
SCA + 2.5 ppm AgTANPs	76.61 ± 11.99	69.64 ± 20.95	26.12 ± 53.77	39.52 ± 12.25
SCA + 5 ppm AgTANPs	93.25 ± 5.51	70.22 ± 23.66	30.21 ± 30.65	91.88 ± 4.68
SCA + 10 ppm AgTANPs	96.86 ± 0.47	70.51 ± 8.57	89.24 ± 6.75	86.60 ± 14.37
O-F	no activity	4.28 ± 20.95	no activity	36.52 ± 27
O-F + 1.25 ppm AgTANPs	no activity	60.86 ± 31.18	no activity	33.56 ± 35.88
O-F + 2.5 ppm AgTANPs	88.46 ± 10.27	88.28 ± 9.49	no activity	88.66 ± 3.55
O-F + 5 ppm AgTANPs	83.16 ± 4.94	99.79 ± 0.19	49.16 ± 16.42	97.15 ± 0.39
O-F + 10 ppm AgTANPs	89.24 ± 7.57	100	80.93 ± 4.96	98.83 ± 0.49
ReNu	no activity	14.99 ± 10.53	22.46 ± 4.27	no activity
RenNu + 1.25 ppm AgTANPs	27.82 ± 46.06	34.03 ± 15.06	42.64 ± 15.32	no activity
ReNu + 2.5 ppm AgTANPs	48.99 ± 34.85	38.09 ± 9.06	42.64 ± 5.74	no activity
ReNu + 5 ppm AgTANPs	89.85 ± 0.86	65.58 ± 19.18	48.08 ± 26.44	50.21 ± 7.87
ReNu + 10 ppm AgTANPs	92.99 ± 3.43	82.16 ± 3.06	78.96 ± 7.51	69.77 ± 17.01

## Data Availability

The datasets used and/or analyzed in the current study are available from the corresponding author upon reasonable request.

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
