# Peer review of "Tannic Acid-Modified Silver Nanoparticles in Conjunction with Contact Lens Solutions Are Useful for Progress against the Adhesion of Acanthamoeba spp. to Contact Lenses"

_microorganisms, 2022, doi:10.3390/microorganisms10061076_

Round 1

Reviewer 1 Report

Late diagnosis of Acanthamoeba infection can lead to complete loss of vision. Most patients diagnosed with Acanthamoeba keratitis (AK) are contact lens wearers. Authors demonstrated the anti-adhesion potential of AgTANPs in combination with selected solutions against Acanthamoeba. The manuscript was well prepared.

This is an interesting paper investigating the anti-adhesion effects of tannic acid-modified silver nanoparticles (AgTANPs) in combination with three contact lens solutions against Acanthamoeba spp. on four kinds of contact lenses was determined. The design of the paper is logic and the results are convincing. I have a few questions and comments:

  1. Did the authors count all Acanthamoeba on the entire contact lens or take a random field of photo to count when calculating the number of Acanthamoeba trophozoites adhesion to the contact lens?
  2. Trophozoites were exposed to AgTANPs at different concentrations, and authors drew the conclusion that the same contact lens solutions used in combination with higher centration of AgTANPs showed better anti-adhesive effect. However, AgNPs alone could also reduce the adhesion of amoebae to the contact lens surface. Were AgNPs concentration consistent among groups with different concentrations of AgTANPs?
  3. In Figure 2, it appeared that the morphology of Acanthamoeba changed exposed to high concentrations of AgTANPs. It would be better if the effects of AgTANPs on the morphology and activity of Acanthamoeba was explored in the results.

Reviewer 2 Report

Applying nanoparticles to contact lenses for reduction of amoebic adhesion in terms of prevention of Acanthamoeba infection

This article is submitted as a perspective but should be a research article. Further the novelty is questionable and several amendments are needed in the experimentation and description of the data in this manuscript. This is not a perspective article as methods and results are included. It should be written as a research article. However, basic details in the methods and results are missing.

Abstract: the word cosmopolitan is redundant and should be removed .Firstly, it will be necessary to elaborate the abstract and add information on what actual results were achieved, and a closing statement mentioning how to take the study forward. In the introduction, please mention briefly any recent advances to keratitis therapy to bring the study into context.

Introduction: None of the previous studies on contact lens solution efficacy against Acanthamoeba are mentioned, and there are several. Please add them.

Hussain, R.H.M., Afiqah, W.N., Ghani, M.K.A., Khan, N.A., Siddiqui, R. and Anuar, T.S., 2021. In vitro effects of multi-purpose contact lens disinfecting solutions towards survivability of Acanthamoeba genotype T4 in Malaysia. Saudi journal of biological sciences28(4), pp.2352-2359.

Moon, E.K., Park, H.R., Quan, F.S. and Kong, H.H., 2016. Efficacy of Korean multipurpose contact lens disinfecting solutions against Acanthamoeba castellanii. The Korean journal of parasitology54(6), p.697.

Lakhundi, S., Khan, N.A. and Siddiqui, R., 2014. Inefficacy of marketed contact lens disinfection solutions against keratitis-causing Acanthamoeba castellanii belonging to the T4 genotype. Experimental parasitology141, pp.122-128.

Fears, A.C., Metzinger, R.C., Killeen, S.Z., Reimers, R.S. and Roy, C.J., 2018. Comparative in vitro effectiveness of a novel contact lens multipurpose solution on Acanthamoeba castellanii. Journal of ophthalmic inflammation and infection8(1), pp.1-7.

Abjani, F., Khan, N.A., Yousuf, F.A. and Siddiqui, R., 2016. Targeting cyst wall is an effective strategy in improving the efficacy of marketed contact lens disinfecting solutions against Acanthamoeba castellanii cysts. Contact Lens and Anterior Eye39(3), pp.239-243.

Methods: The nanoparticles must be tested against human cells for cytotoxicity- as this has been done before by the group, it can easily be repeated and the data can be added. Please include these basic experiments. Also is it completely unclear how adhesion assays were performed- please detail the steps clearly and provide references.

Results: How were the results in the table obtained? It is unclear from the methods. Secondly what are the Acanthamoeba adhering to in figure 2? More explanation needs to be made in the methods and results

Discussion.Were the effects on Acanthamoeba cysts determined? If not please add to future studies in the discussion. Also refer to previous studies in this field (citations suggested above). Also discuss the limitations of your study and how you will take this work forward.

Please go through the entire manuscript for grammatical errors.

Reviewer 3 Report

This manuscript describes the use of nanoparticles to inhibit the adhesion of Acanthamoeba to contact lenses. The paper is generally well written and clear.

The title is a little cumbersome and not entirely appropriate since the prevention of Acanthamoeba infection is not addressed in this study. I suggest “Tannic acid-modified silver nanoparticles inhibit the adhesion of Acanthamoeba to contact lenses”

Specific references are required to support the first two sentences of the introduction.

Line 99. While the choice of the Acanthamoeba strain ATCC 30010 is good as many have studied this strain previously, it is also known that this Neff strain, isolated in 1957 and known to be non-pathogenic has adapted to axenic culture and now shows altered properties (Köhsler et al, 2008). Perhaps future studies will be carried out using freshly isolated AK strains known to be pathogenic?

Line 103. It is stated that the Acanthamoeba were cultured with gentamicin at 10mg/mL? This seems to be a very high concentration.  Can the authors confirm that this was the concentration used?

Section 1.6. Although some details of the assay were given such as “washed with saline solution,“  were given in section 1.4 (Line 141), more detail on the amoeba adhesion assay are required so that others can confirm and extend this work.

Köhsler, M., Leitsch, D., Fürnkranz, U., Duchêne, M., Aspöck, H. and Walochnik, J., 2008. Acanthamoeba strains lose their abilities to encyst synchronously upon prolonged axenic culture. Parasitology research102(5), pp.1069-1072.

Round 2

Reviewer 2 Report

Abstract- improved, but some grammatical errors are present and I suggest lines 16-21 can be reduced a bit. Please check the grammar and English.

Introduction- Well written and suggested amendments are included.

Methods-line 148- please add the ATCC number. What formula was used to calculate % cytotoxicity- please include this.

Please justify the selection of contact lens solutions used? Please elaborate briefly on the type of contact lenses used in this study (line 457) and where were they purchased from etc.

Results- There is no characterization data for the nanoparticles. This must be included  or cited it if done previously, as we need to know if the nanoparticles were actually formed (there should be evidence).

Also, it will be good to convert the data into a table as well to show the difference in adhesion.

Discussion- Please justify the use of the particular cell line for cytotoxicity and also the reason for using the particular Acanthamoeba strain. Also, emphasis on the contact lens industry being a billion dollar industry can be added to give the study context (1-2 lines).
